# Low-temperature scattering with the R-matrix method: from nuclear scattering to atom-atom scattering and beyond

Tom Rivlin[1], Laura K. McKemmish[1,2], K. Eryn Spinlove[1] and Jonathan Tennyson[1]⋆

**1** Department of Physics and Astronomy, University College London, London, UK
**2** Department of Chemistry, University of New South Wales, Sydney, Australia
⋆ j.tennyson@ucl.ac.uk

December 20, 2019

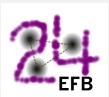

*Proceedings for the 24th edition of European Few Body Conference,*
*Surrey, UK, 2-4 September 2019*

## Abstract

**The R-matrix method is a fully-quantum, time-independent scattering method, used to simulate nuclear, electron-atom, electron-ion, electron-molecule scattering and more. Here, a novel R-matrix method, RmatReact, is presented as applied to ultracold (sub-Kelvin) atom-atom scattering. This is in response to experimental methods which in recent decades have facilitated the routine use of ultracold atoms and small molecules in experiments. This project lays the groundwork for future codes to eventually simulate polyatom-polyatom reactive collisions. Results are presented in this paper for numerical comparisons to other established methods, and for resonances in the elastic scattering of ultracold argon atoms.**

# 1  Introduction

Cold and ultracold atoms and molecules have never been more accessible to working physicists than they are today [1], thanks to a variety of experimental techniques which make them easier than ever to produce [1–3], trap [4], and study [5, 6].

When in these ultracold states, the interactions between atoms and molecules can involve a very small number of quantum states, or possibly only a single state. This is, as Stuhl *et al.* say, "perhaps the most elementary study possible of scattering and reaction dynamics" [1]. As a consequence, it is now possible to study in unprecedented detail the fundamentals of chemical reactions [1, 7]. The unique properties atoms and molecules have when cooled down to ultracold temperatures could pave the way to precise control of reaction rates and improved efficiency of chemical reaction processes, as there are fewer states the reactants can be in, making reactions easier to account for.

For a given system, the exact temperature that can be defined as 'ultracold' can vary, but it is usually when the scattering particles interact with a mutual scattering energy of the order of $10^{-6}$ to 1 Kelvin. In ultracold scattering, the particles themselves may have relatively high energy, but interesting ultracold phenomena are still observed when the translational kinetic energy between the particles is at an ultracold level. For example, merged molecular beam experiments involve molecules travelling alongside each other at speeds which are supersonic in the lab frame, but are only hundreds of millikelvin when measured relative to each other [8].

There are many interesting but subtle implications of this new paradigm. The ability to observe reactive scattering between small atoms and molecules in precisely defined states with minimal thermal noise is, in a sense, one of the most powerful tests possible of the fundamentals of the field of chemistry [1]. The vision of being able to perform a chemical reaction and precisely control the start and end quantum states of all the atoms involved is becoming increasingly possible in many circumstances. This emerging field of research is in need of a robust theoretical framework to assist with these experimental endeavours [9].

One of the most important goals in the theoretical study of these systems is accurate prediction of resonances: the key feature of ultracold collisions which make them useful for controlling reactions. This is also one of the most challenging goals. Resonances are features of the cross-section which arise in conjunction with a variety of physical phenomena, such as the presence of quasibound states with non-trivial lifetimes, or the coupling of closed channels with open ones. Resonances can affect the dynamics and scattering observables of the system at certain energies in quantifiable ways [19, 20]. In heavy-particle collisions, these resonances tend to only be distinguishable at very low temperatures [21]. When they appear in plots of scattering observables such as the cross-section, resonances have a number of characteristic functional forms, such as the Breit-Wigner form [10, 22, 23] and the Fano profile [24]. Of particular interest to this work are systems with deep potential wells which can support many bound states, where there may be many opportunities for novel physics to be uncovered [25].

Theoretical studies of ultracold collisions require a different theoretical framework to room temperature or even cold collisions. For these warmer cases, it is often preferable to use time-dependent methods [26], but these methods struggle at ultracold temperatures where collision times are long and, the resonant states have finite lifetimes which are long relative to the interaction time.

By contrast, time-independent methods have no such difficulties, as they treat the problem with no time variable by solving the time-independent Schrödinger equation. Examples of such

methods include a wide variety of coupled-channel methods, including Hutson's MOLSCAT method [27], and different hyperspherical coordinate methods [28–30], including those presented by Kendrick [31] and by Launay *et al.* [32]. However, these methods still have significant problems, particularly when treating systems with deep potential wells, and hence a new approach could prove fruitful.

This paper is designed to contribute to the need for new theoretical approaches by providing a method for generating high-accuracy scattering observables for collisions involving two atoms, building upon our recent results [44, 46, 47]. The new algorithm we are developing, known as **RmatReact**, is based on the calculable R-matrix method, which was first created to simulate nuclear collisions, and which is now widely applied to electron-atom [10], electron-molecule [11] and nuclear [12, 13] collisions.

Here it has been adapted to the calculation of scattering observables in the atom-atom case. Within the standard Born-Oppenheimer approximation, which is the bedrock of molecular physics, there is a separation between electronic and nuclear motions. While the so-called calculable R-matrix method has been very extensively applied to the electron collision problem, with one exception [14] discussed below, this method has not been applied to problems involving nuclear motion.

However, there are many reasons to expect this largely unused approach to be effective in describing ultracold, heavy-particle collisions between strongly interacting systems. The short-range interactions are complicated and require a high-accuracy multi-dimensional treatment, which is provided by variational nuclear motion codes. Meanwhile, the system must be considered to very long range, i.e. propagation may be required over huge distances and numbers of points [15], meaning that it is advantageous to make the long-range problem as simple as possible, something that can naturally be achieved with the calculable R-matrix method.

The calculable R-matrix method should be particularly suitable for collisions over deep potential wells where, even for three atom problems, systems can support a thousand or more bound vibrational states [16, 17].

Treating ultracold heavy-particle collisions in this way requires significant adaptation to any of the previous calculable R-matrix approaches, including the development of new theory, algorithms and numerical procedures. Our initial proposal for RmatReact [46] focused on the theoretical development of this new approach, highlighting in particular the division of space into the inner, outer, and asymptotic regions connected by the R-matrix. More recent papers have presented results for atom-atom collisions over a Morse oscillator [47] and for Ar-Ar scattering [44].

This paper describes further exploratory studies for non-reactive atom-atom collisions. It shows progress in resolving the technical problems using calculations on test systems. In particular, we focus on the differences between the Calculable and Propagator-Based R-matrix methods.

The work is performed in anticipation of studies on more complex reactions, including atom-diatom collisions and beyond. In these cases, it will be possible to study *reactions*, where, for instance, the atom $A$ collides with the diatom $BC$, and the diatom $AB$ and the atom $C$ are produced by the reaction. The beginnings of this are outlined by McKemmish & Tennyson [18]. Eventually, it is hoped that it will be possible to study ultracold collisions between two polyatoms, each with an arbitrary number of atoms.

For the atom-atom collisions being studied here, it is only possible for the two atoms to have their quantum numbers (their 'channels') altered by the process. We do not consider the possibility that the atoms remain bound after the collision, or that one of both of the atoms are ionised.

## 2   The R-Matrix Method

The particular time-independent method proposed in this work is based on R-matrix methods. These methods are established standards for electron-atom and electron-molecule collisions at 'low temperatures' – so-called light particle scattering [10, 11, 33, 34] – where the energy is below the ionisation threshold of $10^5$ K.

The R-matrix in its original form was invented by Wigner and Eisenbud in the 1940s as an entirely quantum mechanical method for solving *nuclear* scattering problems [35, 36] and rapidly developed into a robust formal theory [37]. The initial impetus for the development of the method was the issue in nuclear physics there was no robust theory of nucleon-nucleon interactions and thus no *ab initio* method of modelling short-range interactions. As such, the R-matrix method developed around the idea of partitioning space into an inner and outer region along the reaction coordinate(s). The two regions could then be treated with different levels of approximation model the scattering observables [10]. Wigner and Eisenbud's approach simple, empirical approximations were used for the inner region, and they simply assumed the boundary between the inner and outer regions to be the asymptotic distance, and did not make use of the long-range physics. This original approach can thus be described as the 'Phenomenological' R-Matrix Method.

By contrast, the R-matrix methods commonly in use today can be thought of as 'Calculable' methods, since they utilise well-established theoretical formalisms for the short-range physics to obtain highly-accurate inner region calculations. Indeed in this work the inner region is calculated using a state-of-the-art nuclear motion code called DUO usually used in high-accuracy spectroscopy [38] and well-established potential energy curves (PECs). The outer region of these methods is also more sophisticated, usually inputting the known, long-range interactions into 'propagator' methods to obtain the value of the R-matrix at some distant asymptotic value from the value of the R-matrix on the boundary.

The R-matrix formalism is naturally suited to the type of problem being studied here. There are fundamental differences between the short-range and long-range behaviours of the atoms as they interact: at short range the interaction is complex, and it merits a high-level of approximation to capture all of the physics involved. At long-range the interactions are simpler, yet a larger range of distances must be considered for the physics of the long-range interactions to be fully captured. A potentially large increase in computational efficiency can be obtained from the way the long-range is handled by the R-matrix method.

With the exception of a single proof-of-principle study by Bocchetta and Gerratt [14], the Calculable R-matrix method has not been applied to so-called heavy particle scattering before. Their work is unique because when heavy particle scattering is usually simulated using R-matrix methods, an alternate R-matrix method is used. This alternate method only uses the aforementioned outer-region propagation part of the method [39–41]. As such, one can refer to these methods as Propagator R-Matrix Methods, in contrast to the Calculable methods used here and by Bocchetta and Gerratt, and to the Phenomenological methods used by the progenitors of R-matrix theory.

In Propagator methods, the boundary between the two regions is set to where the colliding atoms/moleculas are close together in a region characterised by a large repulsive potentia. In this classically forbidden region the system's wavefunctions, and by extension the R-matrix, will be vanishingly small, and so it is assumed to be zero. This R-matrix is then propagated from the boundary, through the outer region, over somepotential energy surface to the asymptotic region. This makes this technique especially useful for heavy-particle scattering, where there often is little resonance structure in the scattering observables, and thus not a large need to produce

Table 1: Comparison between the three families of R-matrix methods discussed in this work.

| R-matrix method | Inner Region | Outer Region | Major use |
|---|---|---|---|
| Phenomenological | Empirical | None | Nuclear scattering |
| Calculable | *Ab initio* | Propagation | Light-particle scattering<br>Heavy-particle scattering (this work) |
| Propagator | None | Propagation | Heavy-particle scattering |

fine-grained plots of observables as a function of energy. The Propagator has the advantage that they avoid the need for diagonalisation in the inner region. Since diagonalisation is usually an expensive process, any method which avoids the need for it has many built-in advantages. A comparison between the three types of R-matrix method can be seen in Table 1

It is anticipated that the Calculable R-matrix methods presented in this work will be an improvement on the Propagator method for the systems being considered here, both in terms of computational expense and in the accuracy of the simulation. This is due to one key aspect of the algorithm. Specifically, Propagator methods must repeat the entire calculation for every different scattering energy value one wishes to sample. Therefore they do not leverage the efficiency of variational nuclear motion programs at solving the Schrödinger equation at short range. This is especially problematic if the potential is rapidly varying, as with, for instance the deep potential wells studied here. By contrast, the Calculable R-Matrix Method employed in this work is especially well-suited to the problems arising in the study of systems with deep potential wells. This is due to the fact that the inner-region needs to be solved only once. Deep potential wells tend to have a very large number of vibrational energy levels below dissociation, and in particular they have complicated short-range physics involving many partial waves at small distances. The consequence of this is that a large basis set is needed to account for all of these states in the diagonalisation process. Because the Calculable approach performs this diagonalisation only once (and can discard high-lying states from the R-matrix sum [42]), it has efficiency advantages when compared to Propagator R-Matrix methods

In this work, the inner region was calculated using a version of the nuclear motion code DUO which has been modified to use Lobatto shape functions [43] for the numerical integration that provides the eigenenergies and eigenfunctions for the R-matrix sum, as defined in the following section. This modification is discussed more in Rivlin *et al.* [44]. The outer region was calculated using a custom implementation of the Light-Walker propagation [39].

## 3   RmatReact Theory

Here modern R-matrix theory is presented as applied to the one-dimensional case of two atoms colliding, based on explanations given by Burke [10] and by Tennyson [11, 45]. The RmatReact method is also in the process of being adapted to the atom-diatom case, as seen in McKemmish *et al.* [18], but only atom-atom collisions are considered here.

In the R-matrix method, space is partitioned into an inner and an outer region. The partition is placed at a distance $a_0$ away from the centre of the reaction, i.e. the 0 of the reaction coordinate. There is furthermore a third region, known as the asymptotic region, which begins at a distance $a_p$ away from $r = 0$ such that $a_p > a_0$. The point $a_p$ is defined as the point where the effects of the

potential can be regarded as negligible – it is the numerical version of assuming infinite distance.

## 3.1 The Inner Region

In the inner region, the two-atom system is assumed to be a quasi-bound diatom, and the PEC of the reaction is simply the PEC of a diatom. In the single-channel, elastic scattering case, the Schrödinger equation for a single partial wave labelled $J$ for this diatomic system has the form

$$\hat{H}^J \psi_n^J(r) = E_k^J \psi_n^J(r), \tag{1}$$

where the $\hat{H}^J$ is the Hamiltonian of the diatomic system, and the $\psi_n^J$ and $E_n^J$ are the eigenfunctions and eigenenergies associated with that Hamiltonian labelled by their angular momentum quantum numbers $J$. In the single channel case, the Hamiltonian and eigenfunctions are all scalars.

The core aim of the inner region calculation is to determine these eigenfunctions and eigenenergies, since they are used to construct an R-matrix, as will be seen shortly.

The diatomic, one-dimensional Hamiltonian $\hat{H}$ of interest is defined in the following way:

$$\hat{H}^J = \frac{-\hbar^2}{2\mu} \frac{d^2}{dr^2} + \frac{\hbar^2 J(J+1)}{2\mu r^2} + V(r), \tag{2}$$

where $r$ is the internuclear separation of the diatom, $J$ is its total angular momentum, and $V(r)$ is the potentials associated with the atom–atom interaction. For a given value of $J$, one obtains the eigenenergies and eigenfunctions $E_k^J$ and $\psi_k^J$ of the Hamiltonian in Eq. (2) using the Schrödinger equation of Eq. (1).

The potential energy surfaces (which are just curves in this work, as there is only one reaction coordinate in the form of the internuclear distance) and the coupling strengths can be thought of as the inputs to the entire algorithm. They are obtained from a variety of experimental and theoretical sources.

The implication of the inner region/outer region split is that the inner-region Schrödinger equation is only valid over a finite, bounded region of space: outside of the $[r_{\min}, a_0]$ range, the Hamiltonian described above is not defined. This is arguably the core principle of the R-matrix method.

In principle, the choice of where to place the minimum distance $r_{\min}$ and the R-matrix boundary $a_0$ should not affect the underlying physics. In practice, the choice of where to define $a_0$ has significant computational implications in terms of accuracy and computational expense of the calculation, as discussed in a previous paper on RmatReact [47].

The R-matrix provides the link between the inner region bound state problem of Eq. (1) and the full scattering Schrödinger equation, as shown in the following derivation. When expressed as an operator (divided by $\frac{\hbar^2}{2\mu}$), the full scattering Schrödinger equation has the form

$$L^J(r) = \frac{d^2}{dr^2} - \frac{2\mu}{\hbar^2} V^J(r) + k^2, \tag{3}$$

where $V^J(r)$ is the potential term containing the angular momentum term, and $k$ is the scattering wavenumber in the body-fixed reference frame, related to the scattering energy by the equation

$$k = \frac{\sqrt{2\mu E}}{\hbar}. \tag{4}$$

When this operator is applied to the full scattering wavefunction, the iime-independent Schrödinger equation is obtained:

$$L^J(r)\psi^J(r) = 0. \tag{5}$$

The derivation of the link between Eq. (3) and Eq. (2) begins by defining another operator:

$$\mathcal{L}_{a_0} = \delta(r - a_0)\frac{d}{dr}, \tag{6}$$

where $\delta(r - a_0)$ is a Dirac delta centred around $a_0$. This operator is a quantity known as the *Bloch* operator [48], and $\mathcal{L}_{a_0}\psi_k^J(r)$ is a quantity known as the *surface term*.

The operator $L^J(r)$ is not Hermitian when integrated over a finite region of space, but the operator $\left(L^J(r) - \mathcal{L}_{a_0}\right)$ is Hermitian. This is because the surface terms left over when integrating over the operator $L^J(r)$ are precisely cancelled by the terms introduced by the operator $\mathcal{L}_{a_0}$. Note that if one integrates over an infinite region of space, and assumes that at infinite distances the functions $v$ and $w$ tend to zero, then $L^J$ is Hermitian, as one would expect. It is the combination of a finite boundary and non-zero boundary conditions at that boundary that introduce non-Hermiticity to the problem.

The operator $\left(L^J(r) - \mathcal{L}_{a_0}\right)$ has a complete spectrum of eigenvalues and eigenvectors over the range $r = 0$ to $r = a_0$. This means that the Green's function [49] of the operator can be represented in terms of those eigenvalues and eigenvectors, due to the completeness relation. It can be shown that the R-matrix *is* the spectral representation of the Green's function of the operator $\left(L^J(r) - \mathcal{L}_{a_0}\right)$.

By subtracting $\mathcal{L}_{a_0}\psi^J(r)$ from both sides of Eq. (5), one can see that the equation

$$\left(L^J(r) - \mathcal{L}_{a_0}\right)\psi^J(r) = -\mathcal{L}_{a_0}\psi^J(r), \tag{7}$$

has the formal solution

$$\psi^J(r) = -\left(L^J(r) - \mathcal{L}_{a_0}\right)^{-1}\mathcal{L}_{a_0}\psi^J(r). \tag{8}$$

If the eigenvalues of $\left(L^J(r) - \mathcal{L}_{a_0}\right)$ are defined to be $\frac{2\mu}{\hbar^2}\left(E - E_k^J\right)$, and the eigenvectors are defined to be $\chi_k^J$, then the Green's function of the operator can be written as

$$G^J(r, r') = \frac{\hbar^2}{2\mu}\sum_{k=1}^{\infty}\frac{\chi_k^J(r)\chi_k^J(r')}{E - E_k^J}. \tag{9}$$

Hence, one can obtain the formal solution $\psi^J(r)$ of Eq. (8) via the equation

$$\psi^J(r) = -\int_0^{a_0} G^J(r, r')\mathcal{L}_{a_0}\psi^J(r')dr', \tag{10}$$

which, due to the Dirac delta in the definition of $\mathcal{L}_{a_0}$, results in the expression

$$\psi^J(r) = \frac{\hbar^2}{2\mu}\left(\sum_{k=1}^{\infty}\frac{\chi_k^J(r)\chi_k^J(a_0)}{E_k^J - E}\right)\frac{d\psi^J}{dr}\bigg|_{r=a_0}. \tag{11}$$

By evaluating this expression at $a_0$, and multiplying and dividing by $a_0$, the expression

$$R^J(E, a_0) = \frac{\hbar^2}{2\mu a_0}\sum_{i=k}^{\infty}\frac{\left(\chi_k^J(a_0)\right)^2}{E_k^J - E} \tag{12}$$

can be identified with the quantity previously defined as the R-matrix itself. The R-matrix is the Green's function of the operator $\left(L^J(r) - \mathcal{L}_{a_0}\right)$ evaluated at the point $a_0$. Hence it is possible to write

$$\psi^J(a_0) = a_0 R^J(E, a_0) \frac{d\psi^J}{dr}\bigg|_{r=a_0}. \tag{13}$$

Equation (13) shows that the R-matrix can be thought of as the log-derivative of the scattering wavefunction $\psi^J(r)$, evaluated at the inner region boundary $a_0$. Furthermore, the R-matrix is constructed of the eigenvectors and eigenvalues of the operator $\left(L^J(r) - \mathcal{L}_{a_0}\right)$, which suggests a clear method for calculating this quantity based on a calculation of the eigenvalues and eigenvectors of the Hamiltonian constrained to the inner region.

The computational advantages are also apparent: the computation to obtain the eigenvalues and eigenvectors only needs to be performed once, and then the R-matrix can be computed for a number of scattering energies $E$ at no extra cost.

## 3.2 The Outer Region

In the outer region, the R-matrix is propagated from the inner region boundary $a_0$ to an asymptotic distance $a_p$. In this work, the Light-Walker method [10, 39] is used to perform the propagation, although other methods exist, too. In the single-channel case, this propagator is simply an iteration equation, that uses the PEC to produce the value of the R-matrix at a given scattering energy $E$ and distance $a_s$, $R_s(E)$, from the value of the R-matrix at the same scattering energy and a shorter distance $R_{s-1}(E)$. The scattering energy and the value of the PEC at the point $a_s$ are contained within the quantity $\lambda_s$, which is defined as

$$\lambda_s^2 = k^2 - \frac{2\mu}{\hbar^2} V^J(a_s) \tag{14}$$

(where the quantity $V^J(a_s)$ contains the angular momentum component of the potential).

The iteration procedure can then be shown to take the form

$$R_s(E) = \frac{-1}{a_s \lambda_s(E)} \left( \frac{1}{\tan(\lambda_s(E)\delta a)} + \frac{2}{\sin(2\lambda_s(E)\delta a)} \left(a_{s-1} R_{s-1}(E) \lambda_s(E) \tan(\lambda_s(E)\delta a) - 1\right)^{-1} \right), \tag{15}$$

where $R_s(E)$ is the value of the R-matrix at each iteration step for a given energy $E$ (beginning at $R_{a_0}$), and $\delta_a = a_s - a_{s-1}$.

The scattering observables can be obtained from the value of the R-matrix at $a_0$. In other words, it is not strictly necessary to have an outer region. However, since it is assumed that the PEC is zero at the asymptotic distance where scattering observables are calculated, it is clearly preferable to evaluate these observables at as far a distance as possible.

In the single-channel case, the K-matrix is given in terms of modified spherical Bessel and Neumann functions $s_J(x)$ and $c_J(x)$:

$$s_\nu(x) = x j_\nu(x) \tag{16}$$

$$c_\nu(x) = -x n_\nu(x), \tag{17}$$

where $j_\nu(x)$ is a spherical Bessel function of the first kind and $n_\nu(x)$ is a spherical Neumann function (or a spherical Bessel function of the second kind) [50]. Using these definitions, the K-matrix takes the form

$$K^J(k) = \frac{-s_J(ka_p) - R^J(E, a_p)ka_p s_J'(ka_p)}{c_J(ka_p) - R^J(E, a_p)ka_p c_J'(ka_p)}, \tag{18}$$

where $s'_J(x)$ and $c'_J(x)$ are the respective derivatives with respect to $x$ of $s_J(x)$ and $c_J(x)$.

The arctangent of the K-matrix can be defined to be a quantity called the phase shift, or *eigenphase*. This is not an observable quantity itself, but observables such as the cross-section and the scattering length can be defined from it, and features such as resonances can be seen in it. As such, many results in the following section are presented in the form of eigenphases. The eigenphase is defined for a given partial wave, and the total cross-section can be obtained by summing over contributions from individual partial waves.

Resonances that appear in the eigenphase can be modelled and fitted using the functional form of Breit and Wigner [10, 22, 23]:

$$\delta^J(E) = A_0 + A_1 E + \arctan \frac{\Gamma_{\text{res}}}{E - E_{\text{res}}}, \tag{19}$$

where $A_0$ and $A_1$ are simple fitting parameters that linearly model the background eigenphase, $\delta^J(E)$ is the eigenphase for partial wave $J$ as a function of the scattering energy $E$, $\Gamma_{\text{res}}$ is the resonance width, and $E_{\text{res}}$ is the location of the resonance. The definition of $\Gamma_{\text{res}}$ differs by factor of two from the definition of the full width at half maximum (FWHM) of a function, and is the same as the convention used by, for instance, Tennyson & Noble [23].

### 3.3 Multichannel Theory

Although only single-channel results are presented in this work, for completeness the multichannel formulation of the inner region part of the method is presented here. The key difference between the single and multichannel cases is the replacement of the PEC $V(r)$ with a matrix of PECs and couplings, $\mathbf{V}(r)$. The elements of the matrix of potentials, $V_{ii'}(r)$, correspond to the elements of the R-matrix itself. The possibility for the scattering event to cause transition between atomic channels is captured by different elements of the R-matrix.

Much of the theory concerning the R-matrix which this discussion is derived from is written from the perspective of electron-atom and electron-molecule scattering [10, 11, 45]. For heavy particle scattering this must be reformulated to account for reduced mass terms and possibly different reference frames. However when the heavy particles themselves have structure, *i.e.* for collisions between open shell species, further considerable differences arise which will be discussed elsewhere [51].

Some fundamentals remain the same between the single and multichannel definitions, and the different systems. There is still a time-independent Schrödinger equation:

$$\mathbf{H}^\Gamma \mathbf{\Psi}_k^\Gamma = E_k^\Gamma \mathbf{\Psi}_k^\Gamma, \tag{20}$$

except now the Hamiltonian $\mathbf{H}^\Gamma$ is a matrix quantity, and the scattering wavefunction $\mathbf{\Psi}_k^\Gamma$ is a vector quantity, where the $N_c$ elements of the vector $\mathbf{\Psi}_k^\Gamma$ correspond to the $N_c$ different channels $i$ of the interaction. The Hamiltonian contains the potential matrix, along with the matrix of kinetic operators. The specific form of the kinetic matrix will depend on the reference frame one is in.

Here, $k$ labels the different eigenfunctions of the scattering Hamiltonian, and $\Gamma$ labels the symmetry of the scattering event by listing all the good quantum numbers. In this work, $\Gamma$ represents $J$ and $p$, the total angular momentum, and the parity of the interaction respectively. Splitting the interaction by symmetry like this is a form of partial wave expansion, which must be consolidated when measuring certain scattering observables.

From here the same derivation as in the single channel case can be performed. The Bloch operator becomes a diagonal matrix with the surface term at $a_0$ for each channel on the diagonal,

and again a Green's function can be found for the operator formed from subtracting the Bloch operator from the Hamiltonian. This Green's function can then be identified again with the R-matrix, such that one can form the equation

$$F_i^\Gamma(a_0) = \frac{\hbar^2}{2\mu} \left( \sum_{i'=1}^{N} \sum_{k=1}^{\infty} \frac{w_{ik}^\Gamma(a_0) w_{i'k}^\Gamma(a_0)}{E_k^\Gamma - E} \right) \frac{dF_{i'}^\Gamma}{dr} \bigg|_{r=a_0}. \tag{21}$$

In this equation, several new quantities have been introduced. The quantity $F_i^\Gamma(r)$ is the *channel function* for a given channel $i$ in a given symmetry $\Gamma$ – the radial wave function in the outer region for a given channel. Eq. (21) shows that this quantity is, in general, dependent on contributions from other channels, and one must sum over all the solutions of the scattering Schrödinger equation to obtain it.

The quantity $w_{ik}^\Gamma(a_0)$ is called the *surface amplitude* for a given channel $i$, solution $k$, and symmetry $\Gamma$, at the inner region boundary $a_0$. It can be thought of as the contribution from a single channel to the scattering solution $k$. Eq. (21) suggests the following definition of an element of the R-matrix which couples channel $i$ to channel $i'$:

$$R_{ii'}^\Gamma(E, a_0) = \frac{\hbar^2}{2\mu a_0} \sum_{k=1}^{\infty} \frac{w_{ik}^\Gamma(a_0) w_{i'k}^\Gamma(a_0)}{E_k^\Gamma - E}, \tag{22}$$

which suggests that Equation 21 can be written in the form

$$F_i^\Gamma(a_0) = a_0 \sum_{i'=1}^{N} R_{ii'}^\Gamma(E, a_0) \frac{dF_{i'}^\Gamma}{dr} \bigg|_{r=a_0}. \tag{23}$$

Equations (22) and (23) are analogous to the two R-matrix definitions introduced in the last section. The former is a definition based solely on the scattering energy and inner region quantities, and the latter suggests that the R-matrix can be thought of as the log-derivative of the channel function, which is an outer region quantity. This once again demonstrates the R-matrix's role as a connection between the two regions. Note that the R-matrix is symmetric.

## 4 Results

### 4.1 Numerical Testing

In order to evaluate the effectiveness of the Calculable R-Matrix Method as applied to single-channel, heavy-particle scattering, the Calculable method was tested using different numerical parameters, and it was tested against different R-matrix methods. In particular, the R-matrix method was tested both with and without an inner region, with and without a propagator, and, for the methods which used an inner region, a large and a small inner region.

The particular system studied was the case of an argon atom elastically scattering off an argon atom of the same isotope (A = 40), with several different PECs, as elaborated on further in Rivlin *et al.* [44].

In order to obtain an error metric to compare different methods, the single-channel eigenphase was calculated at 1,000 energies between 0.005 cm$^{-1}$ and 0.5 cm$^{-1}$(1 eV $\sim$ 8065 cm$^{-1}$), and the

root-mean-square-deviation, or RMSD, $\delta_{\text{RMSD}}(E)$ of the eigenphases calculated at these values using different methods was obtained:

$$\delta_{\text{RMSD}} = \sqrt{\frac{\sum_{i=1}^{1000}(\delta_{\text{ana}}(E_i) - \delta_{\text{num}}(E_i))^2}{1000}}, \tag{24}$$

where $\delta_{\text{A}}(E_i)$ and $\delta_{\text{B}}(E_i)$ are the eigenphase at the energy $E_i$ calculated using method A and method B respectively. All eigenphases, and the RMSD, are in radians.

The results of these comparisons are presented in Table 2. In each row, two methods are compared. Methods with an inner region and a propagator are the Calculable R-Matrix Methods explored in this work, methods with a propagator but no inner region are the Propagator-Based R-Matrix Methods and, for completeness, methods with an inner region but no propagator are also presented – in these methods, the K-matrix is evaluated at $a_0$, and not some asymptotic distance $a_p$.

The 'Small' inner region was one where the Hamiltonian of Eq. (1) was defined from $r_{\text{min}} = 2.5$ Å to $a_0 = 12.5$ Å, and the numerical integration was performed with 400 grid points using Lobatto quadrature. The 'Big' inner region was one where the integration was performed from $r_{\text{min}} = 2.5$ Å to $a_0 = 82.5$ Å with 1600 grid points. Although the grid spacing is not consistent, previous research has shown that these grid spacings are sufficiently small enough to not affect the numerics [47].

In the cases that used both an inner region and a propagator, the point $a_p$ was defined to be twice the value of $a_0$: 25 Å for the Small inner region, and 165 Å for the Big inner region. The same number of iterations – 4,000 – was used in both cases.

In the case where only a propagator was used, the point $a_0$ was set to be 0.01 Å, and the point $a_p$ was defined to be 25 Å, and again 4,000 iterations were used. Some testing was done on varying $a_p$ and the number of iterations for this method (following on from results presented in Rivlin *et al.* [47]), and the differences were found to be non-negligible, but they did not qualitatively change the value of $\delta_{\text{RMSD}}$.

The case of a small inner region without a propagator is not presented in Table 2. This is because these results appeared to be qualitatively different to the results generated using the other methods. It is reasonable to conclude that an insufficiently large inner region, combined with a lack of a propagation method, results in incorrect scattering observables being obtained.

Several different $Ar_2$ PECs [52–54] were tested in the inner and outer regions of the method, and similar numerical results were found for each of them. The results in Table 2 were generated using the PEC of Aziz [52]. In both the inner region and the propagator, the full PEC was used. This is in contrast to conventional R-matrix methods, which only use a long range version of the potential in the propagator that only has terms which are polynomial in $r^{-1}$. Again numerical testing was performed which compared using the full potential to only using the long-range form in the propagator, and in this case the differences were found to be negligible.

The computational expense of these methods is not compared here, as these methods are all inexpensive in the single-dimension, single-channel case being considered here, even when a large number of energy and distance grid points are used. This will be a more important issue to consider in future work on the RmatReact method for polyatomic systems.

In principle, the difference between the Big and Small inner regions should be an efficiency issue only. In practice, previous research has shown [47] that placing the boundary $a_0$ insufficiently far out affects the physics of the problem. The Small inner region boundary was placed sufficiently far out that this problem was not encountered, as demonstrated by the fact that there

Table 2: Root mean square deviations between methods with different inner regions and with or without a propagator.

| Method A | | Method B | | $\delta_{\mathrm{RMSD}}$ for $\delta_{\mathrm{A}}$ vs $\delta_{\mathrm{B}}$ (radians) |
|---|---|---|---|---|
| Inner region | Propagator | Inner region | Propagator | |
| Small | Yes | Big | No | $9.92 \times 10^{-3}$ |
| Small | Yes | Big | Yes | $9.94 \times 10^{-3}$ |
| Small | Yes | None | Yes | $1.75 \times 10^{-3}$ |
| Big | No | Big | Yes | $2.65 \times 10^{-5}$ |
| Big | No | None | Yes | $8.69 \times 10^{-3}$ |
| Big | Yes | None | Yes | $8.71 \times 10^{-3}$ |

is no qualitative difference between the eigenphases generated with Small and Big inner regions, provided a propagator is used, too.

As Table 2 shows, there is little quantitative difference between the single-channel, single-dimension results generated using the Calculable R-Matrix Method and the Propagator-Based R-Matrix Method analysed here. The results that are most representative of the method presented here working optimally are the ones produced using the Big inner region with a propagator. Table 2 shows that these results do not differ substantially from the ones generated only using a propagator.

## 4.2 Argon-Argon Scattering

The results presented in this subsection extend the recent results of Rivlin *et al.* [44].

The elastic scattering of an argon atom off another argon atom was simulated at sub-Kelvin temperatures by obtaining eigenphases and cross-sections for various values of total angular momentum $J$. Several $Ar_2$ PECs were tested, but the results in this section are all produced using the PEC of Myatt *et al.* [55].

These results were generated using an inner region that ranged from $r_{\mathrm{min}} = 2.5$ Å to $a_0 = 22.5$ Å, with 500 grid points. The propagation was from $a_0$ to an asymptotic distance of $a_p = 45$ Å, with 4000 iteration steps.

Figure 1 shows a selection of eigenphases and cross-sections for different partial waves. The $J = 0$ cross-section displays the large growth in amplitude typical of low-energy cross-sections. As these plots show, the cross-section can vary over a large number of orders of magnitude at low energy for different values of $J$.

Figure 2 shows a close-up of an eigenphase that does have a resonance – the $J = 10$ eigenphase close to 0.44 cm$^{-1}$. Also featured in this plot is the function obtained by fitting the data produced in this method to the Breit-Wigner functional form of Eq. (19), using the Levenberg-Marquardt algorithm from Origin (OriginLab, Northhampton, MA). The position of this resonance agrees with the position stated in the supplementary material of Myatt *et al.* [55], as stated in [44]. The figure gives values for $A_0$ and $A_1$, the parameters used to linearly approximate the background eigenphase. Tests were also done using one and three background fitting parameters, assuming the background was constant and quadratic respectively, and the position and width of the resonance was found to be the same in both cases. The exact values of $A_0$ and $A_1$ presented here are not to be considered precise, and are only included for completeness.

Figure 3 shows the total cross-section obtained by summing over the partial cross-sections with

even values of $J$ from $J = 0$ to $J = 10$. Only even values of $J$ are allowed by the Pauli principle as $^{40}$Ar, the isotope of argon considered here, is a boson with zero nuclear spin meaning that states of $^{40}$Ar$_2$ with odd $J$ are disallowed.

Two of the resonances which were detected in this work are clearly visible in the structure of Figure 3. One resonance, in the $J = 9$ partial wave, is not visible as it is too narrow. These resonance positions are corroborated by the ones cited in Myatt *et al.* [55]. Again the low-energy growth is clearly visible in the structure of the total cross-section, owing to the contribution from the $J = 0$ partial wave.

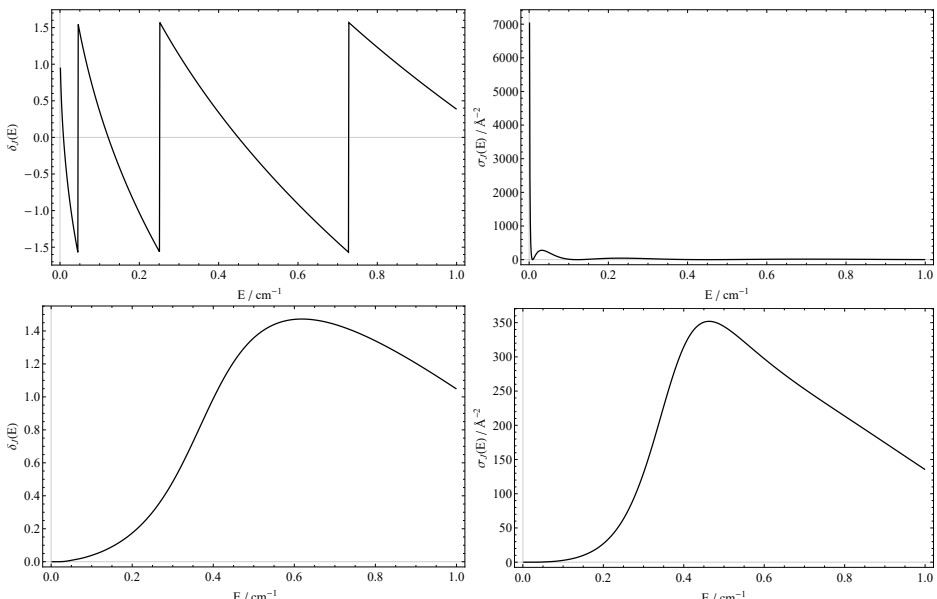

Figure 1: Eigephases (left) and cross-sections (right) for the $J = 0$ (top) and $J = 8$ (bottom) partial waves. The eigenphases are in radians and the cross-sections are in Å$^{-2}$. None of these have resonances – the discontinuities are due to the plots in radians being made between $-\frac{\pi}{2}$ and $\frac{\pi}{2}$.

## 5  Conclusions

Results are presented here for elastic scattering of argon atoms off other argon atoms at ultracold temperatures, using an R-matrix method called RmatReact. Numerical comparisons are presented to other R-matrix methods, and between different sets of numerical parameters within a method. These results demonstrate that the methods used here are comparable to others with respect to accuracy.

The eigenphases and cross-sections produced here are clearly capable of detecting resonances which have been detected in other works, and are thus able to produce observable parameters which could be of use to experimental physicists wishing to study these ultracold systems.

Results generated using this method will soon be presented for the case of multichannel, or inelastic scattering, of atoms. Work has already begun on adapting these methods to the atom-diatom case, too [18], and work is continuing on that topic. Eventually it is hoped that the RmatReact method will be able to simulate the various interesting quantum phenomena that occur at

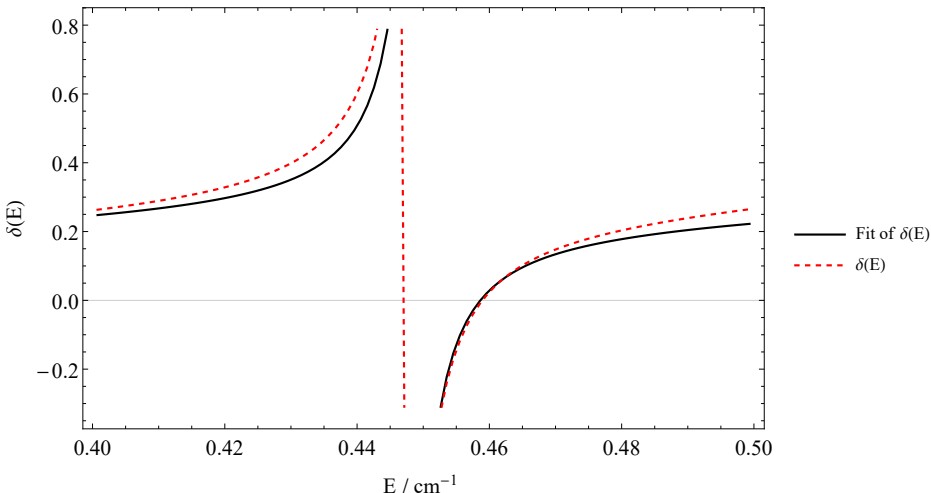

Figure 2: The eigenphase for $J = 10$, along with the Breit-Wigner fit used to model the resonance. The values of the Breit-Wigner parameters obtained from this fit are $A_0 = -0.107$, $A_1 = 0.757$ $(\text{cm}^{-1})^{-1}$, $E_{\text{res}} = 0.4486$ $\text{cm}^{-1}$, $\Gamma_{\text{res}} = 0.00247$ $\text{cm}^{-1}$.

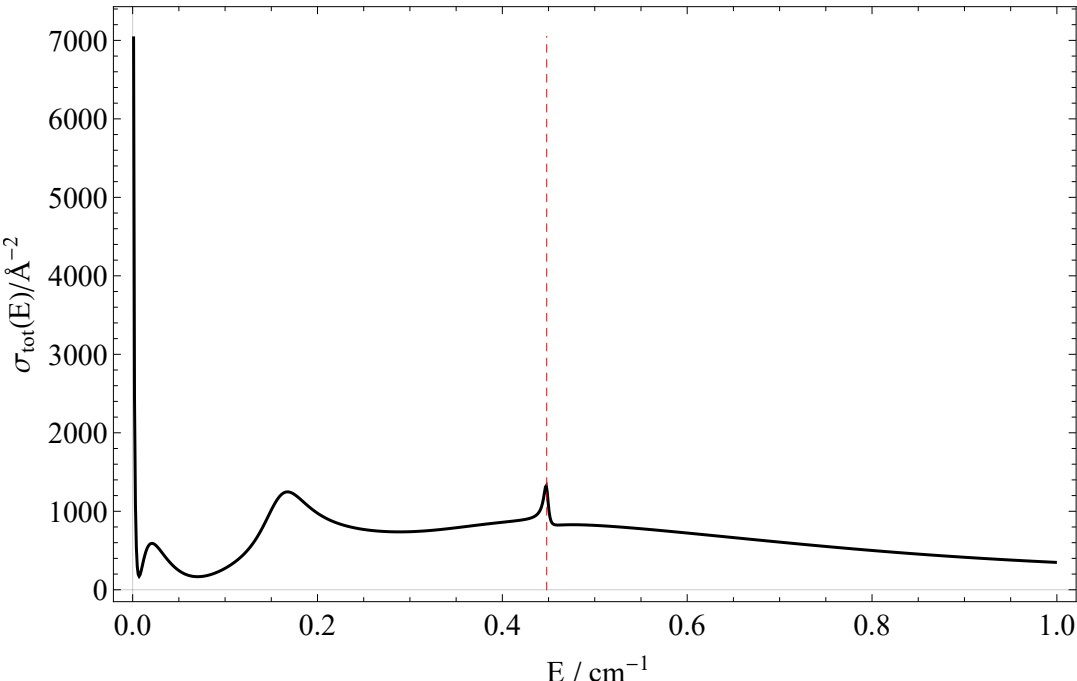

Figure 3: The full cross-section generated by summing the partial cross-sections for even $J$ with $J \leq 10$. Resonance positions are highlighted. The plot also features the cross-section when only summing over even values of $J$.

ultracold temperatures in collisions between molecules.

**Funding information** This project has received funding from the EPSRC under grants EP/M507970/1 and EP/R029342/1, and the European Union's Horizon 2020 research and innovation programme

through Marie Sklodowska-Curie grant agreement No 701962.

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
