# Peer review of "Low-temperature scattering with the R-matrix method: from nuclear scattering to atom-atom scattering and beyond"

_SciPost Physics Proceedings_

## Round 1 · Referee Report · Anonymous · 2019-11-14

Report
The manuscript presents recent developments on R-matrix calculations, applied to ultracold atom-atom scattering. The authors describe a new R-matrix framework, referred to as the "RmatReact method".
I have no doubt that the physics of ultracold systems is fundamental, and that it opens new perspectives in modern physics. However, as nuclear physicist and frequent user of the R-matrix method, I am rather puzzled by the paper. I do not want to bother the authors with minor details, but this text could be a good opportunity to clarify some technical aspects.
My main concerns are as follows:
1) I am surprised that the R-matrix framework needs a specific adaptation for ultracold systems. To my knowledge, going to lower energies is simple since there are less and less nodes in the wave function. I am therefore wondering why a "novel method" is necessary (and, what is "novel", compared to previous works)?
2) As a general statement, the authors emphasize that "RmatReac" is a new development, with several authors, many related publications, and a strong financial support.
In fact, what seems to be done is to solve a one-dimensional Schrodinger equation for scattering states. Unless the potentials present (very) unusual specificities, this looks rather trivial. It may be a misunderstanding on my side, but I think that the difficulties should be better underlined.
3) I am very surprised by Table 2, where standard deviations are given. It might be that the apparently strong differences between both methods are due to slightly different resonance energies. If this is the case, I do not think that the delta_RMSD are significant. If not, one the methods (at least) is very poor.
4) The numerical testing (sect.4.1) involves 400 points in the 'Small' inner region and 1600 grid points in the 'Big' inner region. Again, these huge numbers of basis functions are difficult to understand. It is really necessary? or is it a sledgehammer to crack a nut?
Additionally, where are iterations (4000!) necessary??
Minor questions/comments:
A) In Eq.(9) it seems that the eigenvalues E^J_k are different from those of eq.(1), although the same notation is used.
B) In eq.(15), a reference would be welcome.
C) The parameterization of the background phase shift (19) is somewhat arbitrary. How would the resonance properties (E_res and Gamma_res in Fig.2) be affected if another parameterization is adopted (e.g. only A_0, or A_0,A_1,A_2)? Are all figures significant?
Author: Jonathan Tennyson on 2019-12-20 [id 691]
(in reply to Report 1 on 2019-11-14)My main concerns are as follows: 1) I am surprised that the R-matrix framework needs a specific adaptation for ultracold systems. To my knowledge, going to lower energies is simple since there are less and less nodes in the wave function. I am therefore wondering why a "novel method" is necessary (and, what is "novel", compared to previous works)?
2) As a general statement, the authors emphasize that "RmatReac" is a new development, with several authors, many related publications, and a strong financial support. In fact, what seems to be done is to solve a one-dimensional Schrodinger equation for scattering states. Unless the potentials present (very) unusual specificities, this looks rather trivial. It may be a misunderstanding on my side, but I think that the difficulties should be better underlined.
3) I am very surprised by Table 2, where standard deviations are given. It might be that the apparently strong differences between both methods are due to slightly different resonance energies. If this is the case, I do not think that the delta_RMSD are significant. If not, one the methods (at least) is very poor.
4) The numerical testing (sect.4.1) involves 400 points in the 'Small' inner region and 1600 grid points in the 'Big' inner region. Again, these huge numbers of basis functions are difficult to understand. It is really necessary? or is it a sledgehammer to crack a nut? Additionally, where are iterations (4000!) necessary??
Minor questions/comments: A) In Eq.(9) it seems that the eigenvalues E^J_k are different from those of eq.(1), although the same notation is used.
B) In eq.(15), a reference would be welcome.
C) The parameterization of the background phase shift (19) is somewhat arbitrary. How would the resonance properties (E_res and Gamma_res in Fig.2) be affected if another parameterization is adopted (e.g. only A_0, or A_0,A_1,A_2)? Are all figures significant?

---

## Editorial Decision

submission_&_refereeing_history